# Comprehensive Bibliometric Analysis of Uremic Toxin Research

**DOI:** 10.3390/toxins17110537

**Published:** 2025-10-30

**Authors:** Marsuki Hardjo, Takuya Wakamatsu, Kazuki Watanabe, Shoko Yamazaki, Suguru Yamamoto

**Affiliations:** Division of Clinical Nephrology and Rheumatology, Kidney Research Center, Graduate School of Medical and Dental Sciences, Niigata University, Niigata 951-8510, Japan; dokterukisensei@gmail.com (M.H.); tkywakamatsu@yahoo.co.jp (T.W.); m11112kw@jichi.ac.jp (K.W.); shochan5420@gmail.com (S.Y.)

**Keywords:** uremic toxins, bibliometric analysis, chronic kidney disease

## Abstract

Uremic toxins accumulate as kidney disease progresses, contributing to diverse systemic disorders. Despite numerous studies, no comprehensive mapping has been performed. In this study, keywords related to uremic toxins were quantitatively analyzed using bibliometric methods to clarify research trends, key molecules, and unresolved challenges. Literature and molecular data on uremic toxins and chronic kidney disease were retrieved from the Web of Science and the European Uremic Toxin Work Group (EUTox) databases. Network, citation burst, and keyword frequency analyses were performed using KeyWords Plus. In total, 3302 articles were identified, showing an increasing trend. Citation burst analysis revealed a growing interest in gut microbiota-related topics (burst strength = 15.21), whereas keyword frequency analysis indicated that indoxyl sulfate (566 articles) and p-cresyl sulfate (537 articles) were the most studied toxins. Toxins such as trimethylamine-N-oxide have gained attention over the past 5 years. Analysis of the EUTox database identified 24 protein-bound uremic toxins; among the 94 toxins with unreported clinical toxicity, 15 molecules, including osteocalcin and quinolinic acid, were investigated in <5 studies. These findings suggest that the gut microbiota and related uremic toxins are current research focuses; however, further investigation of underreported uremic toxins is required to define their clinical significance.

## 1. Introduction

Chronic kidney disease (CKD) affects approximately 850 million people worldwide and is projected to become one of the top five causes of years of life lost by 2040 [1,2]. CKD progression is associated with adverse clinical outcomes, including mortality and cardiovascular disease [3,4]. Patients with end-stage kidney disease (ESKD) undergoing dialysis frequently develop systemic complications that impair survival, activities of daily living, and quality of life [4,5]. One contributing factor is the accumulation of uremic toxins [6], which contribute not only to kidney damage but also to the dysfunction of multiple organs [7]. However, their removal using conventional blood purification therapies in patients with ESKD remains insufficient especially for protein-bound uremic toxins (PBUTs) [8]. Advancing both basic and clinical research on the role of uremic toxins in CKD-related systemic disorders, as well as on potential therapeutic strategies, remains an important challenge for improving clinical outcomes in CKD.

The growing recognition of diverse types of uremic toxins has led to the establishment of the European Uremic Toxin Work Group (EUTox) in 1999 to systematically identify and characterize these toxins [9]. Although the definition of uremic toxins has been debated since the 1970s [10], this initiative resulted in a widely recognized definition and classification introduced in 2003, followed by the creation of the EUTox database in 2012. Uremic toxins are now defined as metabolic waste products that accumulate in the blood due to kidney failure and are categorized into small water-soluble compounds, middle molecules, and protein-bound compounds [11,12]. This classification will be further expanded in 2021 and 2023, emphasizing the multidimensional importance of toxin effects on target organs, clearance technologies, and clinical outcomes [13,14,15].

Despite these advances, several challenges remain. No comprehensive efforts have been made to map emerging trends in uremic toxin research, and a better understanding of the evolving paradigms within the scientific community is necessary to obtain a clearer perspective of the field. Additionally, research remains fragmented, often focusing on specific toxins or organ systems, leading to knowledge gaps and research silos. Most studies also concentrate on well-established uremic toxins and methods for identifying new toxins, whereas emerging or less-characterized toxins receive comparatively little attention.

To address these challenges, we conducted a comprehensive bibliometric keyword analysis of uremic toxins. Bibliometric analysis has become an increasingly popular method in medical research for quantifying and characterizing scientific contributions [16]. By systematically mapping research trends using keyword analysis from the EUTox uremic toxin database, this study categorized uremic toxins into three key groups within the context of CKD pathophysiology: emerging toxins, well-established toxins, and underreported toxins.

## 2. Results

A total of 3302 articles on uremic toxins were retrieved from the WoSCC database, reflecting growing interest in the field since the first publication in 1977 (Figure 1). The annual publication rate steadily increased, with contributions from more than 12,000 authors and more than 100,000 cumulative citations across 812 journals.

### 2.1. Emerging Research Keyword Trend Analysis

Emerging research trends were analyzed using keyword variables with the full counting method, validated using two bibliometric analysis software programs. Analysis using CiteSpace (Figure 2) revealed a sharp increase in publications related to the gut microbiota (burst strength: 15.21), indicating growing attention in this area.

A subsequent analysis using VOSviewer (Figure 3) examined keyword co-occurrence using the full counting method and Lin-log modularity. The results revealed a growing interest in terms such as indoxyl sulfate, p-cresyl sulfate, and gut microbiome between 2018 and 2023, whereas clinical terms such as hemodialysis, hemodiafiltration, and membranes were more prominent between 2012 and 2014.

To further characterize these trends, we applied a fractionalized clustering method using VOSviewer (Figure 4), which confirmed the progression toward increasingly specialized research on gut microbiota.

### 2.2. Uremic Toxin Quantitative Keywords Analysis

A bibliometric keyword analysis specifically targeting uremic toxin-related terms from the EUTox Database requires extensive data cleaning and merging of synonyms using a TXT thesaurus file. We then extracted the data using R Bibliometrix frequency term analysis, selecting terms with a frequency greater than five (Figure 5). Based on this analysis, indoxyl sulfate and p-cresyl sulfate dominate the field, with more than 500 publications for each.

An annual frequency term analysis of the top 10 uremic toxins over time was conducted using R Bibliometrix to assess their evolution, as well as to identify recently emerging toxins (Figure 6). Trimethylamine N-oxide (TMAO) has consistently garnered more than five publications annually over the past 5 years, suggesting its potential emergence as a key research topic.

Our findings were visualized and validated using an alternative software method, VOSviewer’s keyword trend network analysis (Figure 7). These results suggest a recent increase in interest in gut microbiota-related toxins, such as TMAO and kynurenic acid, along with a declining trend in traditional advanced glycation end-product-related compounds, including advanced glycation end products and guanidino methylguanidine.

### 2.3. Underreported Uremic Toxin Analysis

The EUTox database is the most comprehensive and curated resource dedicated specifically to the identification of 130 uremic toxin molecules. These molecules are classified based on their molecular properties (water solubility, intermediate molecules, and protein-bound molecules) and clinical significance. We identified 94 toxins of unknown clinical significance and categorized them according to their molecular properties. Among these, 24 PBUTs had unknown clinical significance (Figure 8), and 15 molecules were reported in fewer than five publications; these included 3-deoxyglucosone, α1-acid glycoprotein, angiogenin, fructoselysine, glyoxal, methylglyoxal, indican, indoxyl-β-D-glucuronide, insulin-like growth factor 1, interleukin-10, N(6)-carboxymethyllysine, osteocalcin, pentosidine, putrescine, and quinolinic acid.

## 3. Discussion

The evolution of uremic toxin research has paralleled advancements in analytical techniques such as high-performance liquid chromatography, mass spectrometry, and liquid chromatography–mass spectrometry. The field is now transitioning into an era of metabolomics, broadening the definition of uremic toxins and enabling the identification of novel compounds [17]. However, an overemphasis on well-established toxins and the increasing exploration of new ones may inadvertently create research silos, hindering a more comprehensive understanding of uremic toxicity. The pathophysiology of uremia is complex and involves intricate interactions among numerous molecules rather than just a few dominant toxins.

Given the complex pathophysiology of uremia, which involves intricate interactions among numerous molecules, a comprehensive, data-driven approach is needed. Unlike previous narrative reviews that primarily summarize well-established findings, our bibliometric analysis provides a systematic overview of the field, quantitatively identifying key trends, research gaps, and emerging therapeutic targets. Scientifically, it offers an objective synthesis that supports future experimental and clinical studies, highlighting underexplored toxins and evolving areas such as the gut microbiome. For patients with CKD, these insights contribute to improved care by guiding research toward interventions that may improve complications and outcomes.

Our bibliometric analysis highlights key performance metrics that illustrate the evolution of the field. The increasing number of publications and citations focusing on uremic toxins underscores their growing importance, and highly cited studies have been shaped by their characterization and classification. Analysis of research fragmentation revealed that indoxyl sulfate plays a critical role in shaping both clinical and basic research; β_2_-microglobulin is central to studies on hemodialysis removal, and gut-derived uremic toxins are increasingly recognized in related research. These metrics not only inform the development of uremic toxin research but also guide future studies.

### 3.1. Well-Established Research Areas of Uremic Toxins

Our scientific mapping provides valuable insights into this established research area by identifying topics that continue to attract attention and those that are receiving less focus. Based on our quantitative keyword analysis, uremic toxins such as p-cresyl sulfate and indoxyl sulfate remain central to the research landscape.

Indoxyl sulfate and p-cresyl sulfate are well-characterized PBUTs. An in vitro uremic model of macrophages demonstrated increased production of inflammatory cytokines, offering important insights into the mechanisms underlying chronic inflammation in CKD [18]. A recent in vitro study suggested the potential role of indoxyl sulfate in reducing bone mineral density, contributing to bone mineral disorders in patients with CKD [19]. Additionally, clinical studies have confirmed that lowering PBUT levels using the oral adsorbent AST-120 can improve outcomes in patients undergoing hemodialysis [20,21]. Interestingly, the elevated cancer risk in CKD patients may be partly driven by the accumulation of uremic toxins, which promote inflammation, oxidative stress, and immune dysregulation [22]. The growing interest in these well-established toxins, not only for their pathophysiological roles, but also for their broader clinical implications, has driven advances in hemodialysis technology [23].

### 3.2. Emerging Uremic Toxin Research Areas

Our keyword trend and network analysis provide valuable insights into emerging topics and guide future research directions. Emerging uremic toxins such as TMAO, along with indoxyl sulfate and p-cresyl sulfate, all of which are dependent on the gut microbiota, have become central to current research trends along with microbiota-related terms. In contrast, leptin, β2-microglobulin, and parathyroid hormone are gradually being overshadowed.

The link between gut dysbiosis and uremic toxins was first emphasized in a 2014 review [24], which was influenced by a pivotal clinical study that reported alterations in the intestinal microbiota of patients with CKD [25]. Subsequent research has expanded these findings, suggesting that microbial metabolism in the gut may contribute to uremic toxicity and inflammation through the proliferation of bacterial families possessing urease, uricase, and enzymes responsible for the formation of indole and p-cresol in patients with CKD [26]. This intricate interplay between microbial metabolism and uremic toxicity led to the concept of the brain–gut–kidney axis, which links gut microbiome alterations to cognitive impairment and hypertension [27].

Elevated plasma and urinary TMAO levels are associated with poor long-term survival and an increased cardiovascular risk [28,29]. Subsequent studies reinforced these findings, and our term frequency analysis over time further suggests a growing body of literature linking TMAO to atherosclerosis, heart failure, and frailty [30,31,32]. Another study also reported an association between TMAO and a decline in the measured glomerular filtration rate, highlighting the need for further research on reducing TMAO production as a potential therapeutic strategy for patients with CKD [33]. With further clinical research, assessing plasma and urinary TMAO levels may provide useful biomarkers for predicting CKD-related systemic complications.

These paradigm shifts in gut microbiome research have led to the direct application of potential therapeutic strategies. One promising approach is the adoption of a plant-based, fiber-rich, low-protein diet that directly influences the gut microbiome, reduces the production of uremic toxins, and may slow CKD progression [34]. Another clinical study investigated the role of probiotics in patients with CKD but found no significant benefits, likely due to the complexity of the gut microbiota [35]. Although some studies have reported conflicting results on the efficacy of probiotics, prebiotics, and synbiotics in managing CKD, a systematic review and meta-analysis concluded that the current evidence remains inconclusive [36].

### 3.3. Underreported Areas of Uremic Toxins

Our bibliometric term frequency analysis identified 15 PBUTs with fewer than five publications, suggesting that PBUTs remain clinically under-recognized and may hold promise for the development of novel therapeutic strategies. This limited number of studies can be attributed to several factors. Many of these molecules act as precursors for more complex and biologically active compounds, which have attracted greater research attention; examples include Maillard reaction precursors, such as 3-deoxyglucosone, fructoselysine, glyoxal, and methylglyoxal; indican, a precursor of indole; and polyamine precursors, such as putrescine. Additionally, some of these toxins are specific forms of well-known compounds, and their relevance may be overshadowed by their parent molecules. For example, indoxyl-β-D-glucuronide is derived from indoxyl sulfate; N(6)-carboxymethyllysine and pentosidine are subtypes of advanced glycation end-products; and interleukin-10, an anti-inflammatory cytokine, is usually studied within the broader context of cytokine signaling.

Despite these explanations, several PBUTs remain understudied with respect to uremic toxicity, even though they are biologically relevant in CKD. Osteocalcin is a bone-derived protein secreted by osteoblasts that plays a key role in bone mineralization [37]; in CKD, altered osteocalcin levels influence vascular health by modulating vascular smooth muscle cell function and calcification [38]. Quinolinic acid has recently gained attention because of its role in neurotoxicity and chronic disease [39]; a recent in vivo metabolomic study demonstrated that it exerts toxic effects on both the brain and kidneys [40]. While the roles of these toxins in other chronic diseases are increasingly being recognized, their function as PBUTs and clearance profiles during dialysis remain underreported. Further studies are warranted to clarify their contribution to uremic toxicity.

## 4. Limitations

To the best of our knowledge, this is the first study to systematically map uremic toxins, with a particular emphasis on the quantitative analysis of individual toxins. However, this study has some limitations. First, our analysis focused on publications containing specific keywords such as “uremic toxin” and “CKD” in the abstract. Many earlier, highly cited, basic research studies did not include these keywords because they concentrated solely on individual molecules. Because of this, our analysis should be regarded as a representative model of the research landscape, rather than a comprehensive or definitive overview. Additionally, this study relied on quantitative bibliometric keyword analysis, which may fail to fully capture certain uremic toxins with short or complex names, multiple synonyms, or a low frequency of appearance in the literature. Despite these limitations, this study provides a valuable framework for identifying research gaps and guiding future investigations into clinically relevant yet underreported uremic toxins.

## 5. Conclusions

Research on uremic toxins has increased in recent years, with a particularly high number of studies focusing on indoxyl sulfate, p-cresyl sulfate, TMAO, and gut microbiota. Several molecules, including osteocalcin and quinolinic acid, have been identified; however, their clinical toxicity in the pathophysiology of CKD remains insufficiently understood. Future research should aim to clarify the clinical impact of less frequently reported molecules as potential uremic toxins.

## 6. Materials and Methods

### 6.1. Data Source and Collection Strategy

An electronic search was conducted to identify uremic toxins listed in the EUTox database, and relevant literature was retrieved from the Web of Science Core Collection (WoSCC, version as of 21 January 2025, Clarivate, Philadelphia, PA, USA). The WoSCC is widely regarded as the standard database for bibliometric studies because of its high quality, peer-reviewed journal indexing, comprehensive coverage, and reliable citation tracking, providing greater consistency and precision than other databases [40]. The search strategy combined synonyms for uremic toxins and CKD to obtain clinically relevant results using the following search terms: Topic Search = (“uremic toxin*” OR “uremic toxicity*” OR “kidney-derived toxin*” OR “renal toxin*” OR “uremic metabolite*” OR “protein-bound uremic toxin*” OR “middle molecule*”) AND (“chronic kidney disease*” OR “dialysis” OR “hemodialysis” OR “peritoneal dialysis” OR “end-stage renal disease*” OR “renal failure*” OR “uremia*”). No time-span restrictions were applied, and the search was conducted on 21 January 2025. All data were retrieved within a single day to minimize the bias arising from daily database updates.

### 6.2. Inclusion, Exclusion, and Screening

Eligible studies included original research and review articles published in English. Conference proceedings, book chapters, early access publications, retracted publications, withdrawn publications, non-English language articles, and unpublished studies were excluded. All documents were independently screened by M.H. and T.W. according to these criteria, and S.Y. (Shoko Yamazaki), S.Y. (Suguru Yamamoto), and K.W. supervised interpretation. After excluding 662 items of gray literature, 104 non-English articles, and 21 unpublished studies, 3302 documents were included in the final dataset following a preferred reporting items for systematic reviews and meta-analyses (PRISMA)-guided systematic search [41] (Figure 9).

The search results were exported as plain-text files, merged into a single dataset, filtered for synonyms, and deduplicated using the R Bibliometrix package (version 4.2.1, R Foundation for Statistical Computing, Vienna, Austria). Data cleaning of terms with different spellings, but identical concepts, was performed using a thesaurus synonym text file.

### 6.3. Bibliometric Analysis and Data Visualization

We conducted a bibliometric analysis specifically designed to address four key questions: what are the performance metrics for uremic toxin research; what are the emerging trends; which toxins are the most widely studied, and which have recently emerged; and which toxins remain under-reported. Advanced bibliometric analyses were conducted using VOSviewer (version 1.6.20, Centre for Science and Technology Studies, Leiden University, The Netherlands) [42], CiteSpace (version 6.3, Drexel University, Philadelphia, PA, USA). R1) [43] and the R bibliometrix package [44].

Performance metrics were analyzed using the R Bibliometrix package using the Biblioshiny interface, and visualizations were generated using VOSviewer. To ensure robustness, emerging trends derived from Keyword Plus were explored using two complementary methods: citation burst analysis with CiteSpace and keyword co-occurrence trend network analysis with VOSviewer.

To classify uremic toxins, we first cleaned the data by removing non-toxin-related keywords and merging synonyms. Frequency term analysis using the biblioshiny interface was then applied to categorize toxins as well-studied or underreported. Temporal trends were examined through annual frequency analysis, and newly emerging toxins were identified using the same approach. Data visualization and result validation were performed using VOSviewer.

## Figures and Tables

**Figure 1 toxins-17-00537-f001:**
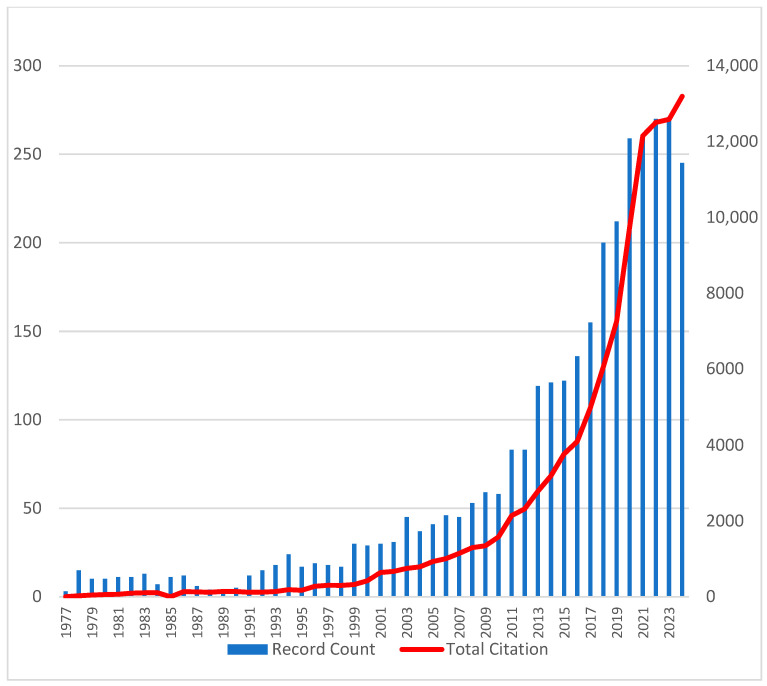
Annual publication number and time cited over time, illustrating the growing impact of uremic toxin research. Blue bars represent the number of publications per year (**left axis**), while the red line indicates the number of citations per year (**right axis**).

**Figure 2 toxins-17-00537-f002:**
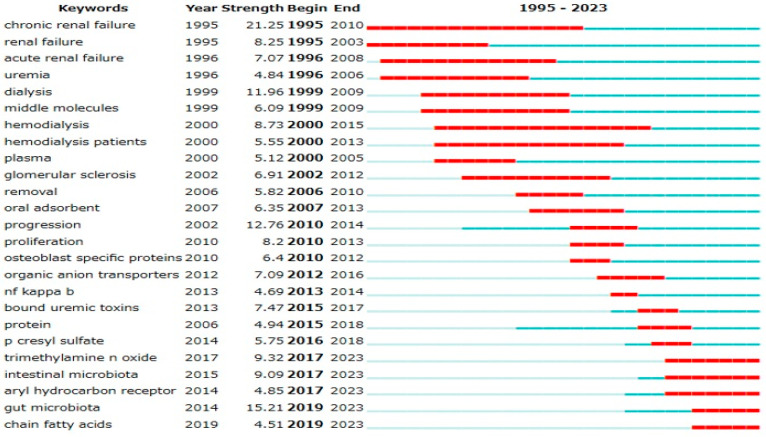
Top 25 keywords with strong citation burst. The red line represents periods of significant citation bursts, while the blue line indicates normal citation periods. The burst strength quantifies the magnitude of the sudden and significant increase in citations by Kleinberg’s Burst Detection Algorithm.

**Figure 3 toxins-17-00537-f003:**
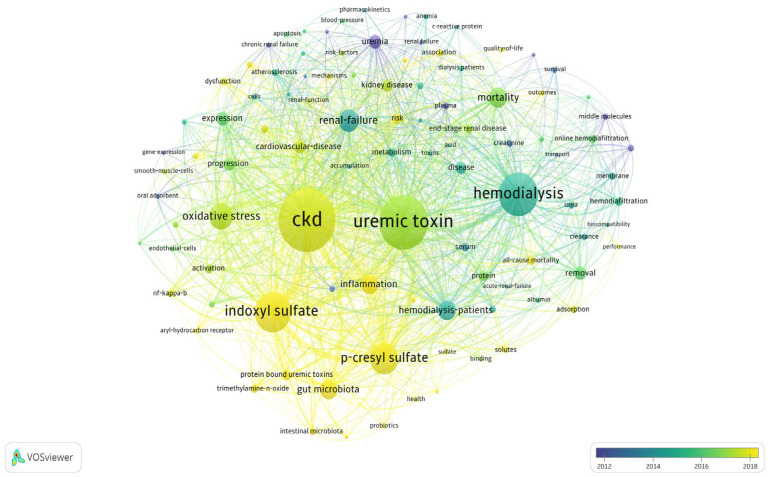
Analysis of keyword co-occurrence trends. Analysis was performed using the full counting method and lin-log modularity, highlighting potential trending keywords in yellow and declining keywords in blue. Circle size represents the quantitative strength of keyword co-occurrence, while line thickness indicates the strength of association.

**Figure 4 toxins-17-00537-f004:**
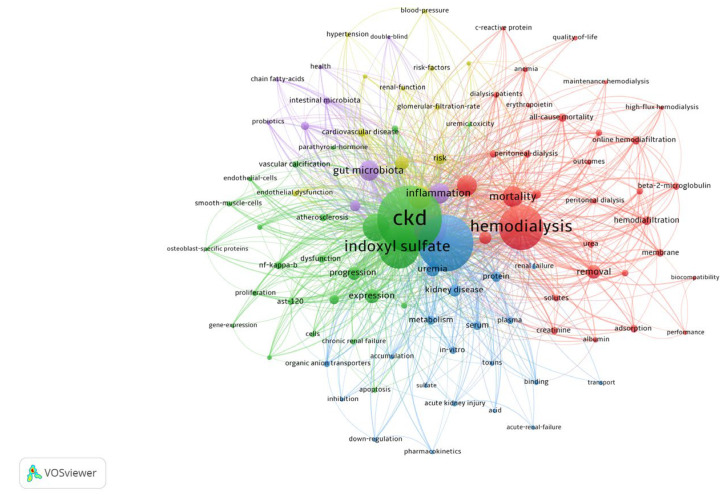
Analysis of keyword co-occurrence clustering. Analysis was performed using the fractionalized method revealed distinct research clusters based on keywords: gut microbiota (purple), clinical studies (yellow), hemodialysis removal (red), functionality of uremic toxins (blue), and wet lab studies (green). These findings highlight the evolving specialization in research, with an increasing focus on gut microbiota-related keywords.

**Figure 5 toxins-17-00537-f005:**
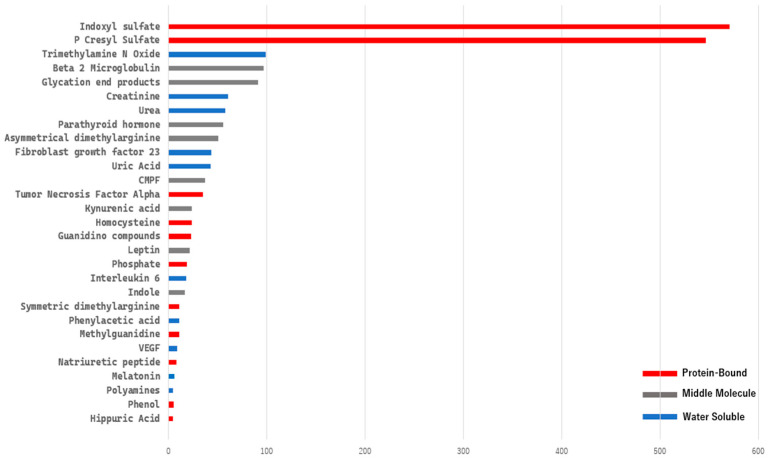
Top 29 keywords associated with uremic toxins, each appearing in more than five publications. Color differences represent toxin classifications: red for protein-bound uremic toxins (PBUTs), blue for water-soluble toxins, and gray for middle molecules.

**Figure 6 toxins-17-00537-f006:**
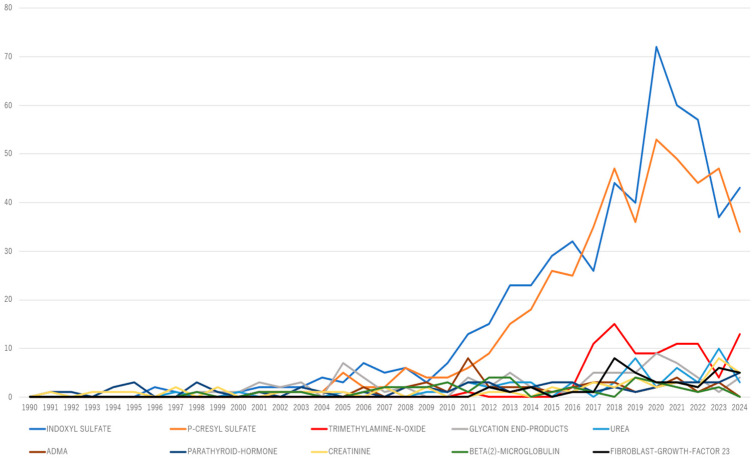
Top 10 uremic toxin keyword co-occurrence trends.

**Figure 7 toxins-17-00537-f007:**
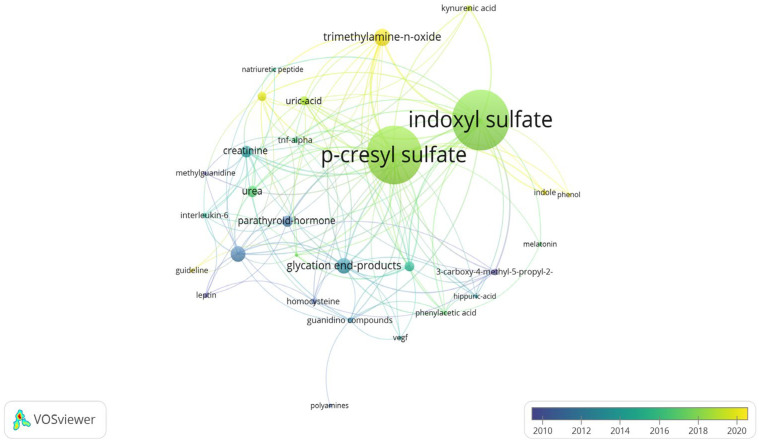
Analysis of keyword co-occurrence trends involving specific uremic toxin keywords. Circle size represents the quantitative strength of keyword co-occurrence, while line thickness indicates the strength of association.

**Figure 8 toxins-17-00537-f008:**
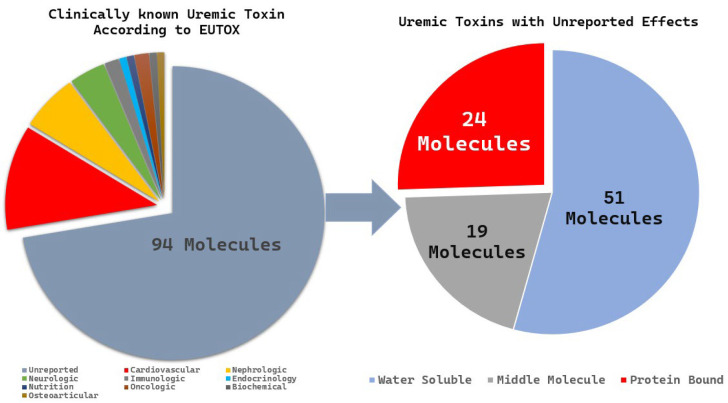
Illustration of the extraction method for uremic toxins, focusing on underreported PBUTs in chronic kidney disease, based on the EUTox database.

**Figure 9 toxins-17-00537-f009:**
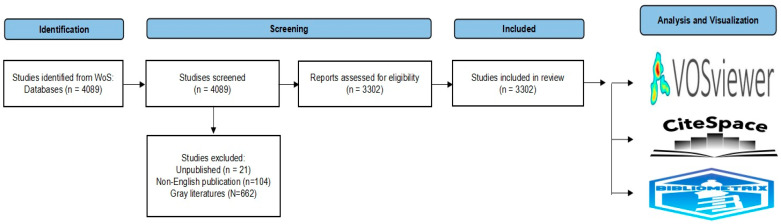
PRISMA-guided systematic search and extraction of 3302 studies.

## Data Availability

The original contributions presented in this study are included in the article. Further inquiries can be directed to the corresponding author.

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
