# Peer review of "Comprehensive Bibliometric Analysis of Uremic Toxin Research"

_toxins, 2025, doi:10.3390/toxins17110537_

Round 1
Reviewer 1 Report
Comments and Suggestions for Authors
The authors have provided a valuable and comprehensive and easily readable summary of contemporary publications on uremic toxins. Limitations are addressed. I found this very informative.
Author Response
We would like to thank the reviewer for the careful evaluation of our manuscript.
Reviewer 2 Report
Comments and Suggestions for Authors
The study is relevant with appropriate research design and methodology, results are clearly presented and background information is solid. However, I have some comments that needs to be addressed.
- What is scientific merit of this article?
- What merit from this article have the patient with CKD?
- This merits should be emphasized in the Discussion part.
Author Response
Comments 1: What is scientific merit of this article?
Response 1: Thank you for pointing this out. The scientific merit of this article lies in its systematic and quantitative approach to understanding uremic toxin research. Unlike traditional narrative reviews, our bibliometric analysis objectively identifies trends, research gaps, and emerging areas in the field. It integrates data from a large body of literature to highlight key molecules, evolving methodologies, and underexplored therapeutic targets. This provides a comprehensive overview that can guide both experimental and clinical research, ensuring future studies are more targeted and scientifically grounded.
Comments 2: What merit from this article have the patient with CKD?
Response 2: Thank you so much for your comment. For patients with CKD, we think that the merit is indirect but significant. By systematically mapping the evolution of uremic toxin research, this study not only highlights well-characterized toxins but also identifies underreported and emerging toxins. This broader perspective can guide the scientific community toward novel therapeutic targets and interventions, potentially preventing complications, optimizing dialysis strategies, and improving patient outcomes. In essence, the study contributes to translational research that may ultimately enhance clinical care for CKD patients.
Comments 3: This merit should be emphasized in the Discussion part.
Response 3: Thank you so much for your comment, and we agree with it. Therefore, we have added a new paragraph in the Discussion section (page 8, lines 199–207).
“Given the complex pathophysiology of uremia, which involves intricate interactions among numerous molecules, a comprehensive, data-driven approach is needed. Unlike previous narrative reviews that primarily summarize well-established findings, our bibliometric analysis provides a systematic overview of the field, quantitatively identifying key trends, research gaps, and emerging therapeutic targets. Scientifically, it offers an objective synthesis that supports future experimental and clinical studies, highlighting underexplored toxins and evolving areas such as the gut microbiome. For patients with CKD, these insights contribute to improved care by guiding research toward interventions that may improve complications and outcomes.” 
Reviewer 3 Report
Comments and Suggestions for Authors
The manuscript presents a comprehensive bibliometric analysis of uremic toxin-related research, integrating data from the Web of Science Core Collection and the EUTox database. The topic is timely and relevant, as the systemic implications of uremic toxins are increasingly recognized with the growth of metabolomics and microbiome studies.
The manuscript is well presented, but the following aspects should be revised by the authors:
- It would be interesting to add the systemic or tumor effects of indoxyl sulfate and p-cresyl sulfate to the discussion. There are several studies in the literature describing the pro-oncogenic effect of these two uremic toxins (doi: 10.3390/genes14061257.). This addition would enrich the discussion by highlighting their broader pathophysiological significance;
- Although the bibliometric analysis is solid, the manuscript should better define its novelty compared to previous reviews. A short paragraph summarizing the specific contribution of the study, such as the identification of poorly explored toxins or evolving trends in the microbiome, would clarify its originality;
- The discussion could also briefly address the clinical relevance of emerging toxins such as TMAO and kynurenic acid. A short paragraph outlining their potential diagnostic or therapeutic implications would strengthen the translational value of the work;
Comments on the Quality of English Language
The language is generally clear, though some sentences are overly long and should be simplified for conciseness.
Author Response
Comments 1: It would be interesting to add the systemic or tumor effects of indoxyl sulfate and p-cresyl sulfate to the discussion. There are several studies in the literature describing the pro-oncogenic effect of these two uremic toxins (doi: 10.3390/genes14061257.). This addition would enrich the discussion by highlighting their broader pathophysiological significance;
Response 1: Thank you so much for your comment. Accordingly, we have added a sentence in the Discussion section (page 9, lines 228–230).
“Interestingly, the elevated cancer risk in CKD patients may be partly driven by the accumulation of uremic toxins, which promote inflammation, oxidative stress, and immune dysregulation [26].” 
Comments 2: Although the bibliometric analysis is solid, the manuscript should better define its novelty compared to previous reviews. A short paragraph summarizing the specific contribution of the study, such as the identification of poorly explored toxins or evolving trends in the microbiome, would clarify its originality;
Response 2: Thank so much for your comment. Accordingly, we have added a paragraph in the Discussion section (page 8, lines 199–207).
“Given the complex pathophysiology of uremia, which involves intricate interactions among numerous molecules, a comprehensive, data-driven approach is needed. Unlike previous narrative reviews that primarily summarize well-established findings, our bibliometric analysis provides a systematic overview of the field, quantitatively identifying key trends, research gaps, and emerging therapeutic targets. Scientifically, it offers an objective synthesis that supports future experimental and clinical studies, highlighting underexplored toxins and evolving areas such as the gut microbiome. For patients with CKD, these insights contribute to improved care by guiding research toward interventions that may improve complications and outcomes.”
Comments 3: The discussion could also briefly address the clinical relevance of emerging toxins such as TMAO and kynurenic acid. A short paragraph outlining their potential diagnostic or therapeutic implications would strengthen the translational value of the work;
Response 3: We thank the reviewer for this suggestion. We respectfully note that the clinical relevance of emerging uremic toxins, including TMAO is already addressed in the Discussion section (page 9, lines 249–255). According to the reviewer’s suggestion, we have added a sentence about future perspectives (page 9, lines 255–257).
“With further clinical research, assessing plasma and urinary TMAO levels may provide useful biomarkers for predicting CKD-related systemic complications.”